# Potential Use of Anti-Inflammatory Synthetic Heparan Sulfate to Attenuate Liver Damage

**DOI:** 10.3390/biomedicines8110503

**Published:** 2020-11-16

**Authors:** Katelyn Arnold, Yi-En Liao, Jian Liu

**Affiliations:** Division of Chemical Biology and Medicinal Chemistry, Eshelman School of Pharmacy, University of North Carolina, Chapel Hill, NC 27599, USA; arnoldk2@email.unc.edu (K.A.); yienliao@live.unc.edu (Y.-E.L.)

**Keywords:** heparan sulfate, chemoenzymatic synthesis, oligosaccharides, DAMPS, inflammation

## Abstract

Heparan sulfate is a highly sulfated polysaccharide abundant on the surface of hepatocytes and surrounding extracellular matrix. Emerging evidence demonstrates that heparan sulfate plays an important role in neutralizing the activities of proinflammatory damage associate molecular patterns (DAMPs) that are released from hepatocytes under pathological conditions. Unlike proteins and nucleic acids, isolation of homogenous heparan sulfate polysaccharides from biological sources is not possible, adding difficulty to study the functional role of heparan sulfate. Recent advancement in the development of a chemoenzymatic approach allows production of a large number of structurally defined oligosaccharides. These oligosaccharides are used to probe the physiological functions of heparan sulfate in liver damage under different pathological conditions. The findings provide a potential new therapeutic agent to treat liver diseases that are associated with excessive inflammation.

## 1. Introduction

Heparan sulfate (HS) is an essential glycan for liver function. Present on the surface of endothelial cells and hepatocytes as well as in the surrounding extracellular matrix, HS participates in regulating blood coagulation, lipoprotein metabolism, cell proliferation, and inflammatory responses [1]. Compared to tools used to study DNA and proteins, the research tools for studying HS are underdeveloped. A multi-year glycoscience initiative from the US National Institutes of Health (NIH) has transformed the research in glycoscience, including HS-related research [2]. The program offers funding support for developing new tools for biologists to conduct glycan-related research topics. This article summarizes the recent development in understanding the roles of HS in liver functions. The primary focus of this article is on the recent progress from our research team on using structurally defined HS oligosaccharides to treat liver damage under sterile inflammation conditions.

## 2. Heparan Sulfate (HS) and HS Proteoglycans

HS is a member of glycosaminoglycans (GAG) family and is comprised of repeating disaccharide units that make up a long chain sulfated polysaccharide. HS is presented as part of proteoglycans of which HS chains are covalently bound to a core protein. HS proteoglycans decorate the cell surface and are present in extracellular matrix (ECM). Cell surface HS proteoglycans include syndecans and glypicans, and they serve as a physical barrier for cells (Figure 1). Syndecan-1 is the primary HS proteoglycan on the surface of hepatocytes. HS proteoglycans have multiple biological functions, including cell adhesion and migration, tissue differentiation, development, and metabolism [3]. It is widely accepted that the biological functions of HS proteoglycan are primarily attributed to its HS polysaccharide chains, whereas core proteins present the HS chain at the right locations and right time to interact with protein effectors. HS plays critical physiological roles as receptors or co-receptors for various proteins. For example, HS on hepatocytes binds to triglyceride-rich lipoproteins (TCL), such as apo-B and apo-E, to mediate their endocytosis and clearance [4]. Reducing *N*-sulfation on hepatocyte HS significantly increases accumulation of TCL even in the absence of LDL receptor, a main clearance receptor for TCL. Also, hepatic HS is found to regulate iron homeostasis by acting as a hepcidin receptor, a liver-derived peptide hormone responsible for iron export [5,6]. Exogenous highly sulfated HS, such as a mixture of ~17 saccharides with 2/6-*O*-sulfation, inhibits hepcidin expression [7].

Under pathological conditions, such as acute liver failure [8,9], membrane-bound HS is released into the ECM and the bloodstream, a process known as proteoglycan shedding. The shedding process occurs when the core protein part of HS proteoglycan is cleaved by proteases [10]. The shed proteoglycans can be further degraded by heparanase, an enzyme that digests long HS chains into small oligosaccharide fragments. These HS fragments are important mediators of inflammation [11], chemotaxis [12], coagulation [13], and infection [14,15] by interacting with signaling molecules (Figure 2). Circulating HS fragments are found to facilitate ECM reconstitution, attenuate inflammatory response by binding to fibroblast growth factor (FGF) and fibroblast growth factor receptor (FGFR) and neutralize damage associated molecular patterns (DAMPs) such as histones [16,17]. Heparin is a highly sulfated form of HS and is released by activated mast cells during inflammation and/or infection to increase vascular permeability. Heparin attenuates coagulation by interaction with antithrombin, thrombin, and factor Xa [13]. HS structures and sulfation patterns are different between normal and pathological conditions. For instance, more 3-*O*-sulfated and 6-*O*-desulfated HS are found in the liver with fibrogenic liver diseases and hepatocellular cancer [18]. However, the significance of the change in HS structure in relation to its function remain unknown. One key question for HS-related research is to identify which specific saccharide sequence relates to biological functions.

## 3. Biosynthesis and Chemoenzymatic Synthesis of HS

### 3.1. Biosynthesis of HS

In contrast to DNA, RNA, and proteins, HS synthesis is not a template-driven process. HS isolated from cells or tissues are a complex mixture of polysaccharides with different sulfation patterns and chain lengths due to the lack of control of the biosynthesis of HS [19]. HS biosynthesis involves multiple biosynthetic HS enzymes. The process can be categorized into linkage formation, polymerization, epimerization and sulfation [20] (Figure 3). First, xylosyltransferases (XylT) initiate the linkage formation by transferring a xylose residue from UDP-xylose (uridinediphospho xylose) to their attachment sites on core proteins. Usually, the attachment site is a serine residue next to glycine surrounded by several acidic amino acids [21]. Two galactose residues and one glucuronic acid residue are added after xylose by galactosyltransferases (GalT-I and GalT-II) and glucuronosyltransferase (GlcAT-I), forming a tetrasaccharide as an initial linkage for HS (Figure 3C1). Second, an *N*-acetylglucosamine is attached to the tetrasaccharide at the non-reducing end by *N*-acetylglucosaminyltransferase I and II (GlcNAcT-I and GlcNAcT-II) to begin the polymerization of HS chains. The chain is elongated by HS glucuronytransferase II and *N*-acteylglucurotransferase II (HSGlcAT-II and GlcNAcT-II) by adding repeating disaccharide units of glucuronic acid (GlcA) and glucosamine (GlcNAc). GlcA can be altered to iduronic acid (IdoA) by glucuronyl C_5_-epimerase (C_5_-epi). The GlcA residue in HS is present in *^4^C_1_*-chair conformation, whereas the conformation of IdoA is more flexible, displaying both *^1^C_4_*-chair conformation and *^2^S_0_*-skew boat conformation (Figure 3C2). Therefore, the conversion of a GlcA residue to an IdoA residue renders the conformational flexibility to HS [22]. Lastly, HS chains can be diversified by sulfation at the 2-position of GlcA/IdoA and/or *N*-, 3-, 6- position of glucosamine by *N*-deacetylase/*N*-sulfotransferase (NDST), 2-/3-/6-*O*-sulfotransferases (2-OST, 3-OST, 6-OST). It should be noted that epimerization and sulfation are ordered processes; *N*-sulfation is required before epimerization reaction, and 2-*O*-sulfation occurs before 6-*O*-sulfation [23]. The substrate specificities of epimerase, 2-OST and 6-OST have limited the structural randomness in the final HS products.

### 3.2. Chemoenzymatic Synthesis of HS

Chemical synthesis of structurally defined HS is difficult [24]. The availability of HS oligosaccharides has been a major limiting factor to advance HS-research. The understanding of the HS biosynthesis and the availability of the recombinant HS biosynthetic enzymes provide an alternative approach to synthesize HS oligosaccharides using the chemoenzymatic approach [25]. The method can diversify the structures of HS in many aspects, including HS length, degree, and pattern of sulfation. Compared to chemical synthesis, chemoenzymatic synthesis shortens the synthetic route and significantly improves the reaction efficiency. Chemoenzymatic synthesis readily generates oligosaccharides that are longer than hexasaccharide. Such capability is particularly important for biological studies as many HS-mediated biological processes require the size of HS fragment larger than six sugar residues (Table 1). Moreover, the variability of HS is further expanded by three 6-OST and seven 3-OST isoforms found in different mammalian tissues. Using these isoforms allows for the possibility of generating tissue-specific HS patterns [20,26]. The method has been licensed to Glycan Therapeutics, LLC to conduct customized reagent synthesis for biological studies. More than 100 different HS oligosaccharides have been synthesized using this method, and these oligosaccharides are commercially available from Glycan Therapeutics (www.glycantherapeutics.com).

The centerpiece in the chemoenzymatic synthesis is the use of recombinant HS biosynthetic enzymes, uridine diphosphate-monosaccharide donors and the sulfo donor for sulfotransferases (Figure 3C3). In order to carry out milligram to gram scale synthesis for biological studies, a large quantity of enzymes and the cofactors, including uridinedisphospho glucuronic acid (UDP-GlcA), uridinediphospho *N*-trifluoroacetyl glucosamine (UDP-GlcNTFA) and 3’-Phosphoadenosine-5’-phosphosulfate (PAPS), is necessary [42]. The expansion of enzyme and cofactor production has led to generation of oligosaccharides with well documented biological activity in animal models [8,43]. Heptasaccharides obtained from this method have been used for crystallogical studies and reveal the complex interaction with 3-OST and 2-OST, indicating that HS with the desired specific structure is available to elucidate sequence specificity between HS and proteins [44,45].

The availability of structurally defined oligosaccharides propels the development of HS microarray technology, a technique that is capable of screening for binding of a protein to different HS oligosaccharides in a high throughput fashion (Table 1). For example, the HS-fibroblast growth factor (FGF)-FGF receptor (FGFR) complex formation and HS binding specificity of FGF1 and FGF2 are further elucidated with chemoenzymatically synthesized HS [32,46]. Moreover, critical pro-inflammatory cytokine and regulator, IL-12 and Wnt are found to bind to HS that has a minimum of octasaccharide domain containing 3-*O* and 6-*O* sulfation, respectively [34,39]. A critical receptor for heparin/LMWH clearance in the liver, stabilin-2, is also shown to have significantly higher affinity to 3-*O*-sulfated HS by applying the customized HS [47]. In summary, the power of chemoenzymatically synthesized HS clearly advances our understanding of the structure–activity and structure–function relationships between HS and their ligands.

## 4. The Potentials for HS-Based Therapeutics

### 4.1. Heparin

Heparin and HS have key structural and functional differences. Heparin is comprised of more IdoA and higher contents of sulfated GlcN. HS contains 0.6 sulfo groups per disaccharide repeating unit with 80% of the disaccharides have a GlcA residue and only 20% of the disaccharides have an IdoA residue [42]. In comparison, heparin contains about 2.6 sulfo groups per disaccharide with 90% the disaccharides from heparin have an IdoA residue and only 10% of the disaccharide from heparin have a GlcA residue [42]. Heparin also carries a higher degree of 3-*O*-sulfation than HS. These structural differences contribute to heparin’s high anticoagulant activity compared to HS. Furthermore, heparin is an exclusive product from the secretory vesicles in mast cells, whereas HS is found on many cell types, including endothelial cells and hepatocytes. 

### 4.2. Anticoagulation

Heparin is a commonly used anticoagulant drug in the clinic. Three forms of heparin-based therapeutics, including unfractionated heparin, low molecular weight heparin (LMWH), and fondaparinux, are currently available on the market. The anticoagulant activity of heparin is determined by interaction with antithrombin and thrombin. Binding to antithrombin (AT) depends on a specific pentasaccharide sequence, including a critical 3-*O* sulfated GlcN, and induces a conformational change in AT. This promotes the formation AT-factor Xa complex thereby inhibiting the downstream coagulation cascade. Also, heparin chains that are longer than 18 saccharides can simultaneously bind to AT and thrombin, forming an inactivated AT-thrombin complex. Only those saccharide chains longer than 18 residues display the anti-factor IIa (or anti-thrombin) activity [48]. LMWH displays high anti-factor Xa activity but low anti-IIa activity because majority of the chains in LMWH are shorter than 18 residues. Fondaparinux only contains the essential pentasaccharide for the binding to AT and is prepared from chemical synthesis. While there are pros and cons for each compound in terms of cost, source, protamine-reversibility, and renal safety, LMWH is increasingly used for venous thrombosis prophylaxis and treatment of deep vein thrombosis in the US and Europe [49].

There are concerns over the reliability and safety of animal-sourced heparin [50]. Unfractionated heparin is isolated from pig intestine through a long and poorly regulated supply chain. LMWH is a depolymerized unfractionated heparin. Therefore, both unfractionated heparin and LWMH are animal-sourced products. A cost-effective method to prepare a heparin substitute using a fully synthetic method will be certainly advantageous. The chemoenzymatic approach is currently the most effective approach to prepare synthetic LMWH. In addition, synthetic LMWH has pharmacological advantages over animal sourced heparin. For example, the anticoagulant activity of synthetic LMWH is fully reversible by protamine, potentially reducing the bleeding side effect associated with currently animal-sourced LMWH [43].

### 4.3. Acetaminophen-Induced Acute Liver Injury

The availability of structurally defined HS oligosaccharides has opened the opportunity to examine potential therapeutic effects in liver disease. The first murine model was acetaminophen-induced acute liver injury. Results from our studies suggest that an 18-mer oligosaccharide is hepatoprotective by neutralizing the proinflammatory activity of high mobility box 1 (HMGB1) [8].

HMGB1, originally called p30 and amphoterin, was first isolated from rat brain using heparin-Sepharose column in 1987 [51]. A few years later, it was discovered that HMGB1 was a ligand for syndecans. The binding between syndecans and HMGB1 was attributed to HS chain [52]. The amino acids in HMGB1 responsible for binding to HS are in the long loop region that connects the A and B boxes. Mutagenesis studies confirmed that Lys-87, Lys-88, Lys-96, Lys-97, and Lys-150 contribute the most to HS binding to HMGB1 [53].

HMGB1 is readily expressed by hepatocytes, more so than other cells types, and therefore has become a subject of interest in many inflammatory liver diseases [54]. Sterile inflammatory liver diseases include alcoholic fatty liver disease, nonalcoholic fatty liver disease, drug-induced liver injury, and ischemia/reperfusion injury during transplantation and liver surgeries [55,56]. HMGB1 expression and translocation from the nucleus to the cytoplasm in human liver tissue from alcohol steatohepatitis patients was increased compared to healthy controls and correlated to disease severity [57]. Using a transgenic mouse with HMGB1 knocked down mainly in hepatocytes, this study also demonstrated that lack of HMGB1 protects from alcoholic fatty liver diseases. The authors attribute this protection to an increase in proteins involved in fatty β-oxidation and a decrease in fatty acid synthase. HMGB1 was also increased in the plasma in a mouse model of nonalcoholic fatty liver disease (NAFLD) induced by high fat diet and use of HMGB1-neutralizing antibody improved liver inflammation although there was no improvement on liver lipid accumulation [58].

Drug induced liver injury caused by acetaminophen (APAP) overdose is the leading cause of acute liver failure (ALF) in many parts of the world [55]. APAP-induced ALF is the best characterized model for HMGB1 dependent injury in patients and animal models. It was the first experimental model system used to show that HMGB1-neutralizing antibody improves the liver condition [54,56]. In an elegant study by Huebener and colleagues, they demonstrate that HMGB1 is necessary for neutrophil migration under sterile inflammation conditions [59]. In vitro, neutrophils significantly migrated towards liver lysate prepared from wild type mice, but not toward liver lyase in which HMGB1 was deleted in hepatocytes and biliary epithelial cells. In vivo, injection of liver lyase containing HMGB1 to the peritoneal cavity induced significant neutrophil migration to the peritoneum. In contrast, injection of HMGB1-depleted liver lyase reduced neutrophil infiltration. Consistent with these results, a reduction in hepatic neutrophil infiltration after APAP overdose in vivo was observed in mice with HMGB1-deficient hepatocytes without altering hepatic macrophages. Importantly, this study demonstrated that the receptor for advanced glycation end products (RAGE), not toll-like receptor 4 (TLR4), is necessary for inflammation after APAP overdose by using TLR4 and RAGE deficient mice.

We discovered an HS oligosaccharide that inhibits the proinflammatory activity of HMGB1 and attenuate liver damage. HS octadecasaccharide (comp1, 18-mer HP, Table 2) protects from ALF caused by APAP overdose [8]. The 18-mer HP was synthesized through the chemoenzymatic approach. The compound contains 18 saccharide residues with a disaccharide repeating unit of -GlcNS-IdoA2S- (Table 2). In this study, we demonstrated that 18-mer-HP’s protective effect is attributed to inhibiting function of HMGB1/RAGE axis and attenuates neutrophil infiltration in the liver (Figure 4). Three lines of evidence supported this conclusion. First, we showed that 18-mer-HP decreased HMGB1-mediated neutrophil infiltration in an air pouch model. Here, we demonstrate that injection of purified recombinant HMGB1 elicits neutrophils into the air pouch while co-injection of 18-mer-HP and HMGB1 reduces the neutrophil infiltration. Second, administration of both 18-mer-HP and HMGB1 neutralizing antibody offered very similar hepatoprotection after APAP overdose. In this experiment, we compared the extent of the hepatoprotection effect from HMGB-neutralizing antibody, 18-mer-HP alone and the combination of 18-mer-HP and HMGB1 antibody in the APAP model. Our findings showed that all three treatments gave the same level of protection, suggesting that both the antibody and 18-mer-HP target to HMGB1. Third, we discovered that 18-mer-HP does not have protective effect in RAGE deficient mice after APAP overdose, suggesting that 18-mer HP is engaged in RAGE/HMGB1 axis to display its hepatoprotection. In a mortality study using a lethal dose of APAP, 18-mer-HP treatment increased the survival rate from 53% to 90%.

Three additional HS oligosaccharides were employed to investigate the contribution of the sulfation patterns and the size of oligosaccharides. These oligosaccharides include comp 4 (6-mer), comp 3 (12-mer), and comp 2 (18-mer AXa). 6-mer and 12-mer have the identical disaccharide repeating structure as 18-mer-HP, but shorter overall size. 18-mer AXa has the same size as 18-mer-HP, but has a higher number of sulfo groups and it has the anticoagulant activity as measured by anti-Factor Xa assay. Neither 6- or 12-mer were able to bind to HMGB1, or decrease liver injury in vivo, suggesting that the size of the oligosaccharide is essential for hepatoprotection. Administration of 18-mer-AXa, although it appears to bind to HMGB1 with higher affinity to 18-mer-HP, does not protect from liver injury. Further studies revealed that the anticoagulant activity from 18-mer AXa reduces fibrin formation, a necessary molecule for liver regeneration after injury. Taken together, the results from the studies uncover a new class of carbohydrates to protect acute liver injury from APAP overdose by targeting to HMGB1.

In addition to 18-mer-HP, endogenous syndecan-1 potentially contributes to the hepatoprotection after APAP overdose. A recent report demonstrated that *Syndecan-1^-/-^* mice were highly sensitive to liver damage caused by APAP toxicity [61]. The authors also demonstrated that purified syndecan-1 and HS polysaccharide, but not the core protein of syndecan-1, rendered protection in *Syndecan-1^-/-^* mice after APAP overdose. Furthermore, they reported that syndecan-1 also displayed protection in APAP overdose wild type mice [61]. The biochemical analysis from our studies revealed that syndecan-1 binds to HMGB1 through its HS side chain. Furthermore, syndecan-1 shedding was readily detected in plasma from both APAP mice and APAP overdose patients [8]. When syndecan-1 is shed from hepatocytes, soluble syndecan-1 may attenuate sterile inflammation by displacing extracellular HMGB1 from the injury sites and/or interacting with circulating HMGB1 (Figure 4). During extensive liver damage, it is likely that shed syndecan-1 is inadequate to neutralize all HMGB1, and the addition of 18-mer-HP, a mimetic of HS on syndecan-1, provides further protection. One plausible hypothesis is that the 18-mer-HP potentiates the host anti-inflammatory effect mediated by syndecan-1.

### 4.4. HS Protects against Liver Damage by Ischemia/Reperfusion Injury

HS oligosaccharides also display protection against liver damage caused by ischemia/reperfusion (I/R) injury. Unlike the case in the APAP model, both anticoagulant and anti-inflammatory activities from a HS oligosaccharide are essential for the protection in liver I/R mouse model. Thrombosis and inflammation are traditionally viewed as separate processes that complement each other, but growing evidence supports the relationship between thrombosis and inflammation stimulating and reinforcing one another especially in sepsis, I/R injury, trauma, and severe burns [62]. Tissue factor lies beneath the endothelium and is exposed during vessel wall injury, where it can serve as a potent activator of coagulation by generating thrombin [62]. Thrombin’s coagulation role entails activation of platelets to a procoagulant state and fibrin generation, while its inflammatory role entails cleavage of C5 in the complement cascade and activation of endothelial cells. Platelets and neutrophils represent the predominant cell types in thromboinflammation [62]. Platelets and neutrophils form heterotypic aggregates on activated endothelial cells, leading to occlusion of the vessel. In addition, activated platelets can also secrete HMGB1 even though they lack a nucleus [63]. Platelets also express HMGB1 receptors, RAGE, TLR2, TLR4, and TLR9, on their surface, suggesting that HMGB1 can stimulate both platelet and neutrophil activation, further propagating thromboinflammation. In liver I/R conditions, hypoxic hepatocytes release HMGB1 [64]. HMGB1 has been shown to recruit neutrophils through RAGE activation after liver I/R injury [59]. Additionally, activation of the complement cascade, specifically thrombin cleavage of completement proteins C3a and C5a, also results in neutrophil recruitment after liver I/R injury. It has been shown that depletion of serum complement before liver ischemia prevents neutrophil accumulation during reperfusion [65]. Once neutrophils infiltrate into the liver, they cause hepatocyte death by releasing protease including elastases, MMP-9, cathepsin G, proteinase-3, and myeloperoxidase (MPO) [66].

We demonstrate that an oligosaccharide with anticoagulant and anti-inflammatory activities protects from liver injury after I/R [60]. Using four 12-mers as well as a 6-mer with different sulfation patterns (comp 3 and comp 5–8, Table 2), we assayed for HMGB1 binding. Two highly sulfated 12-mers (comp 6, 12-mer-NS2S6S and comp 7, 12-mer-AXa, Table 2) bind to HMGB1, suggesting that they possess the anti-inflammatory activity. However, 12-mer-AXa (comp 7) has anticoagulant activity and 12-mer-NS2S6S (comp 6) does not. Both 12-mers were tested in liver I/R injury model. Interestingly, only 12-mer-AXa (comp 7) significantly protects from the injury as determined by plasma concentration of alanine aminotransferase (ALT), percentage of necrotic cells, neutrophil infiltration, and MPO activity in the ischemic liver lobe. We attribute this hepatoprotective effect to 12-mer-AXa’s (comp 7) dual activities of anticoagulation and anti-inflammation as both processes are essential in the pathology of liver I/R injury. To support the hypothesis, we tested a combined treatment of 12-mer-NS2S6S (comp 6) which has anti-inflammation activity but no anticoagulation activity, and comp 8 (6-mer-AXa, Table 2) that has anticoagulation activity but no anti-inflammation effect. As expected, the treatment with the combined oligosaccharides displays protection. The results suggest that both anticoagulation and anti-inflammation are required to protect from I/R liver injury. This can be achieved using a single oligosaccharide, i.e., 12-mer-AXa (comp 7), or a combination of two compounds 12-mer-NS2S6S (comp 6) carrying anti-inflammatory activity and 6-mer-AXa (comp 8) carrying anticoagulant activity.

### 4.5. Other Promising Disease Models for Therapeutic HS

In addition to APAP overdose and I/R injury, heparin/HS are also found to reduce inflammation by lowering pro-inflammatory cytokines and maintaining ECM in CCL4-induced liver fibrosis [67], doxorubicin-induced liver damage [68], and primary graft dysfunction in liver transplantation [69]. Moreover, heparin/HS also have additional therapeutic potential for its antitumor, antivirus, and anti-hepcidin effects; reviews on these topics can be found elsewhere [70,71,72,73]. Briefly, heparin prevents tumor metastasis possibly by reducing adhesion, proliferation, and migration of cancer cells by targeting adhesion molecules (P-selectin, VCAM-1) or chemokines (CXCL12) in preclinical studies [71]. However, the existing data in clinical trials does not support the antimetastatic effect of heparin [74]. Regarding to the antivirus effects, it is shown that heparin/HS can bind to the viral spike proteins to block viral entry and resulting infection, such as herpes simplex virus [75] and hepatitis C virus [76]. Recently, HS proteoglycan is also found to bind to spike proteins of SARS-CoV in vitro [77], which may contribute to beneficial effect of heparin in COVID-19 patients reported [78]. Heparin is shown as a potent inhibitor of hepcidin in hepatic cell model, suggesting its potential therapeutic effects in anemia by restoring iron distribution [79]. In summary, while antitumor, antivirus, and anti-hepcidin effects may also be independent of anticoagulation properties of heparin, structurally defined HS libraries will help to decipher the various roles of HS.

## 5. Conclusions

This review summarizes the recent progress on using HS to reduce liver damage by targeting to HMGB1-mediated sterile inflammation. As a widely used anticoagulant, heparin and its derivatives display a range of therapeutic potentials [80]. Structurally heterogeneous heparin polysaccharides are protective in the APAP overdose model [61,81], but the underlying mechanism has been unclear. The access of structurally homogeneous oligosaccharides enables identification of candidate targets underlying the therapeutic effects. We demonstrate the uses of both anticoagulant and non-anticoagulant oligosaccharides to probe against liver damage in both APAP-overdose and I/R injury models. Effectiveness in the APAP-overdose model requires an oligosaccharide with anti-inflammatory activity and a lack of anticoagulant activity, while effectiveness in the liver I/R model requires both activities. Structurally defined HS oligosaccharides have been perceived as expensive to obtain. The chemoenzymatic synthesis approach has significantly lowered the barrier to access HS oligosaccharides under the support from NIH. The HS-based oligosaccharides will add a new chemical space as therapeutic agents to treat liver diseases.

## Figures and Tables

**Figure 1 biomedicines-08-00503-f001:**
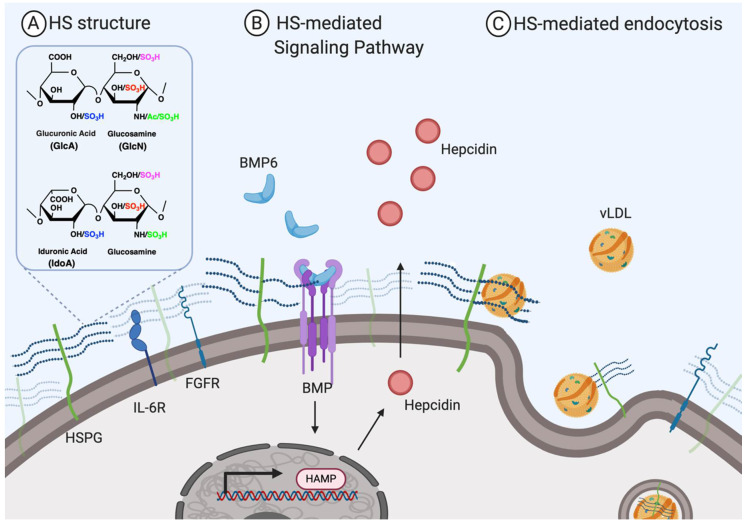
Endogenous heparan sulfate (HS) on the cell surface in a healthy state. (**A**) Membrane-bound HS is usually attached to core protein in the form of HS proteoglycan (HSPG), such as syndecans or glypican. HS is comprised of repeating disaccharides of glucuronic acid (GlcA)-glucosamine (GlcN) and iduronic acid (IdoA)-GlcN in various sulfation pattern as indicated. (**B**) Membrane-bound HS acts as a co-receptor for various ligands, for example, BMP6. In hepatocytes, HS mediates hepcidin expression by modulating BMP6/BMP binding. (**C**) Membrane-bound HS acts as a receptor to facilitate lipid clearance, for example, vLDL in the liver.

**Figure 2 biomedicines-08-00503-f002:**
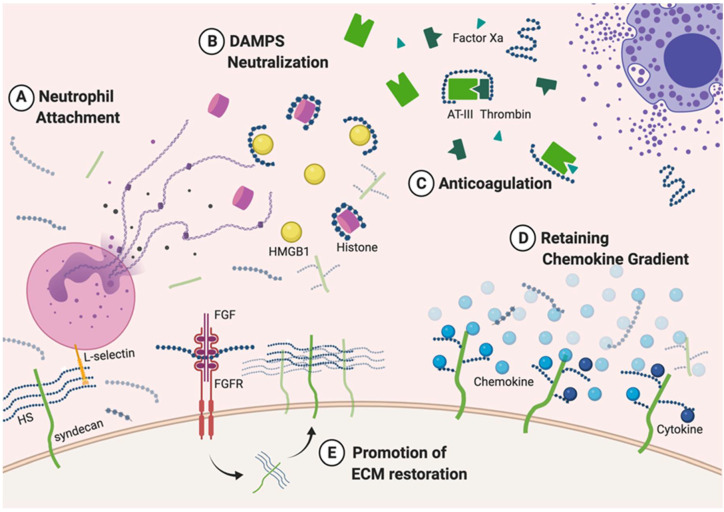
Endogenous HS on the cell surface in a disease state. (**A**) Membrane-bound HS can bind to adhesion molecules on neutrophils, supporting their attachment and rolling on the cell surface. (**B**) Shed HS fragments can bind to damage associate molecular patterns (DAMPs) (for example, histones and HMGB1 release by neutrophils) during inflammation, neutralize them, and prevent DAMPs from potentiating inflammatory response. (**C**) Highly sulfated HS, heparin, is released by activated mast cells and simultaneously binds to antithrombin III (AT-III) and thrombin to inhibit coagulation cascades. Circulating HS fragments can also bind to AT-III and inhibit factor Xa activity; however, they are usually not long enough to inhibit thrombin. (**D**) HS binds to chemokines (ex. CXCL-10, CXCL-12, CCL-2) and cytokines (ex. IL-8, IL-10, IL-12), maintaining their concentration gradients to recruit more immune cells. (**E**) Shed HS fragments act as co-receptors for various ligands, for example, FGF. The formation of HS-FGF-FGFR complex induces downstream signaling pathways for ECM restoration.

**Figure 3 biomedicines-08-00503-f003:**
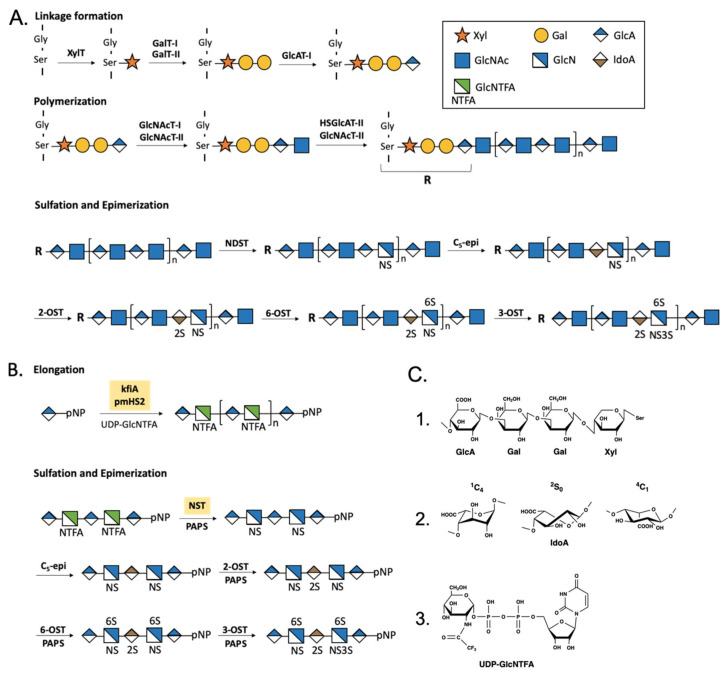
The biosynthesis and chemoenzymatic synthesis of HS. (**A**) The biosynthesis of HS. (**B**) The chemoenzymatic synthesis of HS. Recombinant HS biosynthetic enzymes for polymerization and *N*-sulfation are used to carry out the chemoenzymatic synthesis (highlighted in yellow). Also, unnatural sugar nucleotides such as UDP-GlcNTFA is used in chemoenzymatic synthesis to optimize the synthesis process by increasing specificity and detection. (**C**) The structure of saccharides involved in the synthesis of HS. **1**. Monosaccharides involved in biosynthesis of HS. The structures of GlcA, IdoA, and GlcN can be found in Figure 1A. **2**. The conformation diversity of IdoA. **3**. The structure of unnatural sugar nucleotides used in chemoenzymatic synthesis of HS. kfiA: *N*-acetyl glucosaminyl transferase from *Escherichia coli* K5. pmHS2: heparosan synthase 2 from *Pasteurella multicida*. GlcA-pNP:1-O-(para-nitrophenyl) glucuronide.

**Figure 4 biomedicines-08-00503-f004:**
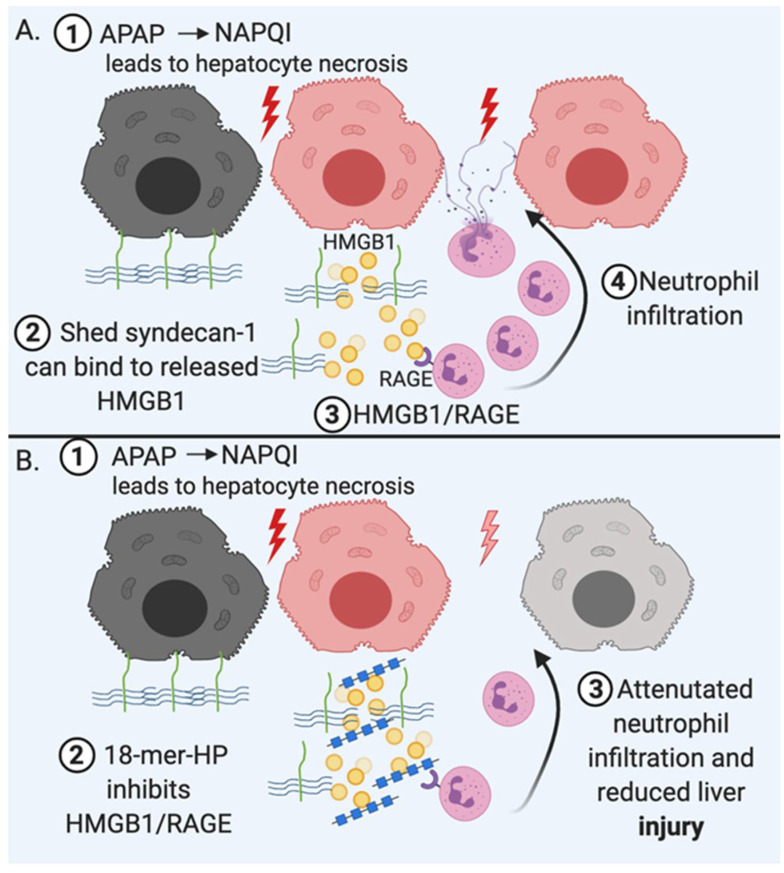
Mechanistic summary of sterile inflammation and 18-mer-HP’s role after acetaminophen (APAP) overdose. (**A**) 1. APAP overdose leads to accumulation of a toxic metabolite, *N*-acetyl-p-benzoquinone imine (NAPQI), which leads to hepatocyte necrosis and subsequent release of damage associated molecular patterns (DAMPS). 2. At the same time cell surface syndecan-1 is shed and can bind to extracellular HMGB1 through is HS chains. 3. HMGB1 that is not neutralized can bind to RAGE, resulting in neutrophil recruitment. 4. Neutrophils propagate initial injury by releasing reactive oxygen species and proteases which act on both damaged and healthy tissue. (**B**) 18-mer-HP reduces liver injury by inhibiting HMGB1-induced neutrophil recruitment.

**Table 1 biomedicines-08-00503-t001:** Binding requirement of HS sequences with protein. HS with specific length and sulfation pattern is required for binding to protein ligands. Numbers in parentheses indicate the length of tested HS, but the minimal length required for binding has not been determined.

Target	Length	Sulfation Pattern of HS	Binding Effects	Ref
**Antithrombin**	5	GlcNS/Ac6S-GlcA-GlcNS3S±6S-IdoA2S-GlcNS6S-	Induce conformational changes, accelerate interaction with factor Xa/thrombin to potentiate anticoagulation	[27,28]
**Thrombin**	>18	--	Simultaneously bind to antithrombin/thrombin, form complexes to potentiate anticoagulation	[29]
**FGFR**	10	6-*O*-sulfation	Form tertiary complex with FGF1 or FGF2	[3,30]
**FGF2**	4	IdoA2S-GlcNS-IdoA2S-GlcNS	Induce dimerization	[31]
	10	IdoA2S-GlcNS6S, terminal GlcNS or GlcNAc	Form tertiary complex with FGF Promote inflammation/induce ECM repair	[32]
**FGF1**	4	IdoA2S-GlcNS6S-IdoA2S-GlcNS6S	Induce dimerization	[33]
	20	IdoA2S-GlcNS6S, terminal GlcNS6S	Form tertiary complex with FGF Reduce JNK-mediated inflammation	[32]
**Wnt**	6	3-*O*-sulfation, 6-*O*-sulfation	Co-receptor for wnt activation Exacerbate liver cancer	[34]
**MCP-1**	6	*N*-sulfation, *O*-sulfationterminal GlcNS6S for 6-mer	Induce oligomerization (tetramer); Retain MCP-1 for leucocyte migration	[35,36]
**IL-8**	8	*N*-sulfation, *O*-sulfation	Mediate IL-8 activity	[37,38]
**IL-12**	8	Highly sulfated, 3S per disaccharide	Stabilize IL-12 and enhance activity	[39]
**Histone**	10	2-*O*-sulfation, N-sulfation	Neutralize histone and reduce inflammation	[17]
**HMGB1**	12	Highly sulfated (NS2S, NS6S, NS2S6S, NS2S3S6S)	Neutralize HMGB1 and reduce inflammation	[8]
**Hepcidin**	>17	*N*-sulfation, 2-*O*-sulfation, 6-*O*-sulfation	Inhibit expression of hepcidin in hepatocytes	[7]
**Neuropilin-1**	(12)	3-*O*-sulfation	Stabilize neuropilin	[40]
**tau**	(12)	3-*O*-sulfation	Inhibit cell surface binding and internalization of tau	[41]

**Table 2 biomedicines-08-00503-t002:** Chemical structures of HS oligosaccharides used in studies [8,60].

Name of Compound	Abbreviated Saccharide Sequence	Reference
Comp 1, 18-mer-HP	GlcNS-GlcA-GlcNS-[IdoA2S-GlcNS]_7_-GlcA-pNP	[8]
Comp 2, 18-mer-AXa	GlcNS6S-GlcA-GlcNS3S6S-[IdoA2S-GlcNS6S]_7_-GlcA-pNP	
Comp 3, 12-mer	GlcNS-GlcA-GlcNS-[IdoA2S-GlcNS]_4_-GlcA-pNP	
Comp 4, 6-mer	GlcNS-GlcA-GlcNS-IdoA2S-GlcNS-GlcA-pNP	
Comp 5, 12-mer-NS6S	GlcNS6S-[GlcA-GlcNS6S]_5_-GlcA-pNP	[60]
Comp 6, 12-mer-NS2S6S	GlcNS6S-GlcA-GlcNS6S-[IdoA2S-GlcNS6S]_4_-GlcA-pNP	
Comp 7, 12-mer-AXa	GlcNS6S-GlcA-GlcNS6S-[IdoA2S-GlcNS6S]_4_-GlcA-pNP	
Comp 8, 6-mer-AXa	GlcNS6S-GlcA-GlcNS3S6S-IdoA2S-GlcNS6S-GlcA-pNP	

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
