# Peer review of "Potential Use of Anti-Inflammatory Synthetic Heparan Sulfate to Attenuate Liver Damage"

_biomedicines, 2020, doi:10.3390/biomedicines8110503_

Round 1
Reviewer 1 Report
This is a very thorough and well prepared review on the exciting topic of the novel therapeutic potential of synthetic heparan sulfates to treat liver damage. The work is timely and of broad interest due to the wider range of potential therapeutic applications of such heparan sulfates. The review is well referenced and supported by helpful and clear figures which help any reader understand the topic well. Only minor revisions are recommended to improve the manuscript.
- Fig 3, panel A on HS biosynthesis should make more clear the longer chain extension for the cartoons of structures modified by sulfotransferases and the epimerase.
- In section 4.3, page 8, the authors need to modify some of the text to past tense in the descriptions of their published work.
- The title needs correction to "Use of...." or "Application of....". And should probably also include the word Potential. So "Potential Use of....".
Author Response
- Fig 3 has been revised according to Reviewer #1's suggestion.
- Description under 4.3 has been changed to past tenses wherever it is appropriate.
- The title has been modified as suggested by th reviewer.
Reviewer 2 Report
In this paper, the authors have reported the recent progress of utilization of chemoenzymatically synthesized heparan sulfate oligosaccharides on inhibition of liver damage by targeting to HMGB1-mediated inflammation. The paper is well written. It can be accepted for publication on the journal. However, several mistakes are observed. They are shown as follows, and should be corrected before accepted.
- Line 34. The second “Heparan Sulfate” should be “HS”, because it has already been abbreviated.
- The structure of GlcA. Although a wavy line is drawn between C5 and C6, it should be a straight line.
- Unify -COOH, -CO2H or -COO-. Carboxy groups are shown as -COO- in figure 1, but as -COOH in Figure 3, and as -CO2H in Figure 4.
- Lines 47 and 165. N-sulfation and O-sulfated are in italic. (Others might be the same somewhere.)
- Line 100 and Figure 3. Glucuronosyltransferase should be abbreviated as GlcAT-I not Gal-AT1.
- Page 3 to 4. The names of genes and those of enzyme activities are mixed up.
Xylosyltransferase (XylT) has two genes, XYLT1 and XYLT2. Galactosyltransferase I (GalT-I) and galactosyltransferase II (GalT-II) are the names for the enzyme activities, and their genes are B4GALT7 and B3GALT6, respectively. The gene name of the enzyme glucuronytransferase I (GlcAT-I) is B3GAT3. EXT1 and EXT2 are the gene names for the enzymatic activities of HS glucuronytransferase II (HSGlcAT-II)/ N-acetylglucosaminyltransferase II (GlcNAcT-II). The enzyme activity of N-acetylglucosaminyltransferase I (GlcNAcT-I) catalyzed by both EXTL2 and EXTL3, but EXTL3 has also the enzyme activity of GlcNAcT-II. The figure should be correctly described.
- Table 1. If there is no limitation on the number of references, the original papers which demonstrated the results shown in the table, should be cited.
- Lines 238-239. “altering hepatic” is doubled.
- Figure 4. It is a little difficult to see the differences among the compounds.
They can also be shown using letters such as follows.
Comp1: GlcNS-GlcA-GlcNS-[IdoA2S-GlcNS]7-GlcA-R
Comp2: GlcNS6S-GlcA-GlcNS3S6S-[IdoA2S-GlcNS6S]7-GlcA-R
- Figure 4, Comp 7. There is an extra rectangule. Explain or remove it.
Author Response
- Line 34, heparan sulfate has been replaced by HS.
- The structure of GlcA in Fig 1 has been corrected.
- We are now using -COOH to represent carboxyl group throughout the manuscript.
- Line 47 and 165, italicized letters are now used. Similar changes are applied in other places throughout the manuscript.
- Line 100, the label for glucuronyltransferase (GlcAT-1) is corrected in Fig 3A.
- We use the names of enzymes, instead of gene names, to depict the biosynthesis of HS in Fig 3A.
- Table 1, references are included.
- Line 238-239, the dupilicated word is deleted.
- Fig 4 is now replaced by Table 2 as recommended by the reviewer.
- This has been corrected in Table 2.
The changes are clearly marked.